# Impact of Glucose-Lowering Medications on Cardiovascular and Metabolic Risk in Type 2 Diabetes

**DOI:** 10.3390/jcm9040912

**Published:** 2020-03-26

**Authors:** Angelo Maria Patti, Ali A Rizvi, Rosaria Vincenza Giglio, Anca Pantea Stoian, Daniela Ligi, Ferdinando Mannello

**Affiliations:** 1Department of Health Promotion, Mother and Child Care, Internal Medicine and Medical Specialties (PROMISE), School of Medicine, University of Palermo, 90121 Palermo, Italy; rosaria.vincenza.giglio@alice.it; 2Division of Endocrinology, Metabolism, and Lipids, Department of Medicine, Emory University School of Medicine, Atlanta, GA 30322, USA; arizvi4@emory.edu; 3Faculty of General Medicine, Diabetes, Nutrition and Metabolic Diseases Department, Carol Davila University, 050474 Bucharest, Romania; ancastoian@yahoo.com; 4Department of Biomolecular Sciences, Section of Biochemistry and Biotechnology, University Carlo Bo of Urbino, 61029 Urbino, Italy; daniela.ligi@uniurb.it

**Keywords:** cardiovascular risk, dipeptidyl peptidase-4 inhibitors, glucagon like peptide-1 receptor agonists, sodium glucose cotransporter-2 inhibitors, type 2 diabetes mellitus

## Abstract

Type 2 Diabetes Mellitus (T2DM) is associated with a high risk of atherosclerotic cardiovascular (CV) disease. Among the well-known pathophysiologic factors, crucial roles are played by endothelial dysfunction (caused by oxidative stress and inflammation hyperglycemia-linked), increased activity of nuclear factor kB, altered macrophage polarization, and reduced synthesis of resident endothelial progenitor cells. As consequence, a potentially rapid progression of the atherosclerotic disease with a higher propensity to unstable plaque is arguable, finally leading to significantly increased cardiovascular mortality. Main managements are focused on both prevention and early diagnosis, by targeted treatment of hyperglycemia and vascular complications. Innovative therapeutic approaches for T2DM seek to customize the antidiabetic treatment to each patient in order to optimize glucose-lowering effects, minimize hypoglycemia and adverse effects, and prevent cardiovascular events. The newer drugs (e.g., Glucagon Like Peptide-1 Receptor Agonists, GLP-1 RAs; Sodium GLucose coTransporter-2 inhibitors, SGLT2is; DiPeptidyl Peptidase-4 inhibitors, and DPP4is) impact body weight, lipid parameters, and blood pressure, as well as endothelial (dys)functions, inflammatory markers, biomarkers of both oxidative stress, and subclinical atherosclerosis. The present review summarizes the results of the main trials focused on the cardiovascular safety of these drugs from the CV standpoint.

## 1. Introduction

Cardiovascular disease (CVD) is the predominant cause of death in diabetic patients [1,2]. Hyperglycaemic status activates multiple maladaptive signalling pathways involving endothelial dysfunction, which leads to the emergence and rapid progression of the atherosclerotic disease with distinct characteristics. The selection of a drug to reduce blood glucose is based not only on its efficacy but also on its cardiovascular safety. Obviously, the prevention and control of CVD in patients with Type 2 Diabetes Mellitus (T2DM) are imperious [3]. Data resulted from recent clinical trials provide updated insightful information about the cardiovascular benefits of therapeutic approaches that are currently available. These trials expanded our knowledge on the efficacy and safety of novel antidiabetic drugs shedding light on undesirable effects about specific aspects of cardiovascular (CV) risk. Diabetic kidney disease (DKD) is the major cause of end-stage renal disease (ESRD). The main therapy that attenuating DKD was the renin-angiotensin system blockade through angiotensin-converting enzyme inhibitors (ACEi) or Angiotensin II Receptor Blocker (ARB) treatments. Currently, it has been suggested as second-line therapy in T2DM patients the use of sodium-glucose transport protein 2 (SGLT2) inhibitors and glucagon-like peptide-1 receptor agonists (GLP-1RAs) for their reno-cardiovascular safety profile [1]. Although they have shown a reassuring safety profile, unfortunately, the current guidelines do not fully allow us to identify the most appropriate drug according to a specific patient phenotype and attendant co-morbidities (Table 1). 

The results of all available CV safety trials facilitate and improve the clinical efforts not only to obtain and maintain an excellent glycemic target, but also to minimize the well-known adverse effects (e.g., weight gain, hypoglycemia, and heart failure).

## 2. Search Strategy

We searched by world-wide electronic scientific databases (i.e., MEDLINE (1975–2019), EMBASE and SCOPUS (2000–2019), DARE (1980–2019)), and Web of Science Core Collection (since 1997) and available abstracts from national and international meetings. The leading search terms were: trials, meta-analyses, Incretins, Receptor antagonists Glucagon-like Peptide-1, inhibitors of DiPeptidyl Peptidase-4, Sodium-glucose transporter-2, ‘kidney disease’ and their association with cardiovascular risk, and prevention of CVD. 

The main and very important Cardiovascular Outcomes Trials (CVOTs) assessing CV safety of glucose-lowering medications reached their primary outcomes and confirming previous studies that indicated that no increased CV risk are detailed in Table 2.

## 3. Traditional Anti-Diabetic Drugs

Metformin is the first choice drug treatment for T2DM; in a UK Prospective Diabetes Study (UKPDS study), when compared with the “conventional” group (i.e., patients treated with diet alone or in combination to chlorpropamide, glibenclamide, or insulin), the series of metformin-treated patients showed significant risk reductions of 32% for any diabetes-related endpoint, 42% for diabetes-related death, and 36% for all-cause mortality in overweight patients [2]. The study named Action in Diabetes and Vascular Disease: Preterax and Diamicron MR Controlled Evaluation (ADVANCE; CT. gov identifier: NCT00949286) highlighted that intensive glycemic treatment based on gliclazide (with modified-release mechanism) reduced the combined endpoint of macro- and microvascular complications, mainly due to the reduction of new nephropathy or worsening of the same [3]. For what concerns thiazolidinediones, the study called Prospective Pioglitazone Clinical Trial in Macrovascular Events (PROactive; CT. gov identifier: NCT00174993), despite the higher incidence of heart failures, revealed that pioglitazone was able to significantly reduce the CV events by about 16% (*p* = 0.027) [4]. 

On the other hand, the trial Thiazolidinediones Or Sulfonylureas Cardiovascular Accidents Intervention (TOSCA; CT. gov identifier: NCT00700856) highlighted that the incidence of CV events was similar between sulfonylureas (i.e., glimepiride and gliclazide) and pioglitazone treatments, even if pioglitazone was strongly associated with a reduced number of hypoglycemic events [5]. No significant difference (1.5 per 100 person-years) was revealed in the primary outcomes (described as the first occurrence of all-cause death, non-fatal myocardial infarction, urgent coronary revascularisation, or non-fatal stroke, assessed in the modified intention-to-treat population) between patients treated with pioglitazone and sulfonylureas (*p* = 0.79). However, in T2DM patients in which metformin alone failed to properly control the glycaemic profile, the best treatment option is actually not fully outlined and remains a matter of strong clinical debate [5]. 

Interestingly, pioglitazone [6] showed after 4.8 years of follow-up a significant reduction of crucial clinical endpoints (MI, fatal or nonfatal stroke) (24%, *p* = 0.02) in 3876 insulin-resistant subjects with a transient or recent ischemic attack or stroke but without diabetes (trial: Insulin Resistance Intervention After Stroke (IRIS; CT. gov identifier: NCT00091949) performed vs. placebo).

In the study Outcome Reduction with an Initial Glargine Intervention (ORIGIN; CT. gov identifier: NCT00069784.), it was reported that, in comparison to standard strategies, the early use and titration of basal insulin did not have any untoward effect on CV events [7]. 

The study named Efficacy and Safety of Degludec versus Glargine in Type 2 Diabetes (DEVOTE; CT.gov identifier: NCT01959529), a trial involving 7637 patients with stated CVD and a mean diabetes duration of 16 years [8], confirmed the CV safety of insulin degludec when compared with insulin glargine. Degludec was in fact statistically superior with a lower rate of both severe and nocturnal severe hypoglycemia (by 40 and 53%, respectively; *p* < 0.001 for both comparisons); no significant differences in CV mortality were found, even if differences in severe hypoglycaemia were also described and discussed [8].

## 4. Novel Anti-Diabetic Drugs

Incretins are well-known intestinal hormones that, after secretion by the entero-endocrine cells in response to oral glucose intake, are able to enhance the insulin secretion from the pancreatic ß-cells. Glucagon-like peptide 1 (GLP-1) and Glucose-dependent insulinotropic peptides (GDIPs) are the two main incretins that are enzymatically degraded in blood by dipeptidyl peptidase-4 (DPP-4). Finally, they are cleared by the renal system [9]. When compared to glucose-dependent insulinotropic peptide, the insulinotropic effect of GLP-1 is more preserved in patients affected by T2DM. 

In recent years, the Incretin-Based Therapies (IBTs) of T2DM (through Glucagon-Like Peptide-1 Receptor Agonists (GLP-1RA), DiPeptidyl Peptidase-4 inhibitors (DPP-4is), and Sodium/GLucose coTransporter 2 inhibitors (SGLT2is)) have been simplified and widely used. As demonstrated by clinical studies, their actions go beyond the glucose-lowering effects demonstrating a plethora of biomolecular pleiotropic effects and targets (i.e., reduction of lipids, blood pressure, inflammatory markers, oxidative stress, endothelial dysfunction, and subclinical atherosclerosis and in body weight). A significantly higher number of T2DM patients showed that sulphonylureas and basal insulin are significantly associated, although not necessarily causatively, with an increasing rate of heart attacks, strokes, and amputations [10]. Accordingly, clinicians could consider more routinely useful the prescription of GLP-1 receptor agonists, SGLT-2 inhibitors, or DPP-4 inhibitors rather than sulfonylureas or basal insulin after metformin, mainly due to the similar/overlapping short-term CV outcomes with GLP-1 receptor agonists, SGLT-2 inhibitors, and DPP-4 inhibitors [10].

### 4.1. Glucagon Like Peptide-1 Receptor Agonists (GLP-1RAs)

GLP-1 improves the glycemic control by enhancing both synthesis and secretion of the glucose-dependent pancreatic insulin pathways, inducing via a paracrine route also the inhibition of glucagon secretion from pancreatic α-cells; moreover, GLP-1 is able to slow down the rate of endogenous glucose production and inhibit gastric emptying promoting finally satiety.

Interestingly, synthetic GLP-1 receptor agonists show high structural similarity to naif GLP-1 but revealed higher resistance to the enzymatic degradation by dipeptidyl peptidase 4 (DDP-4). However, several pieces of data ensured their cardiovascular safety and efficacy. Besides their ability to decrease the levels of blood glucose, GLP-1RAs showed several positive cardiovascular and metabolic effects, such as improved control of blood pressure and cholesterol/dyslipidemia, promotion of weight loss, and reduced food intake, finally reducing the clinical impact of these well-known atherosclerotic risk factors [11]. Furthermore, GLP-1 RAs may positively affect the CV risk through direct actions on the physiology of both myocardium and blood vessels [12]. 

To date, five GLP-1 RAs received FDA approval and include: exenatide, dulaglutide, lixisenatide, semaglutide, and liraglutide [11]. 

Interestingly, beyond the well-known ability of glycemic control, liraglutide has been demonstrated to exert significant effects also in the early stage of atherosclerosis; according to the original hypothesis of Rizzo et al. [13], liraglutide shows in fact potential effects in slowing-down the progression of atherosclerotic disease. Low-Density Lipoproteins (LDL) particles circulating in blood are transported from the vascular space into the arterial endothelial wall, where LDL were crucially transformed in both oxidized and electronegative LDL (ox-LDL). These modified lipoproteins were peculiarly accumulated inside the activated macrophages, significantly contributing to the atherosclerotic plaque initiation inducing inflammatory and proteolytic pathways, well-known early biological-biomolecular events in animal models, in vitro and in vivo atherosclerosis [14]. The significant reduction of ox-LDL clearance by LDL-receptors leads to the specific increase in arterial entry and sub-endothelial retention of harmful ox-LDL, and to their increased in situ peroxidative irreversible modifications. The plethora of bio-molecular events has been proposed and widely accepted as one of the primary mechanisms leading to artery endothelial dysfunctions, subendothelial foam cell accumulation, smooth muscle cell activation, migration, and proliferation in the extracellular matrix, and finally the induction of both platelet activation/adhesion and aggregation. Among these events, liraglutide exerts its potential cardiovascular protective mechanism through direct effects on both plaque initiation/formation and progression (exhaustively reviewed in [13]).

The trial named Liraglutide Effect and Action in Diabetes: Evaluation of Cardiovascular Outcome Results-A Long Term Evaluation (LEADER; CT. gov identifier: NCT01179048) clearly demonstrated in T2DM patients at elevated CV risk that liraglutide (1.8 mg) significantly reduces the rates of major adverse CV events [15]. In a prospective study performed on subjects with T2DM but without CAD [16], Carotid Intima-Media Thickness (CIMT) decreased independently to the well-known effects of liraglutide on both glucose and lipids [17]. Moreover, further studies demonstrated that liraglutide reduced CIMT also in subjects with metabolic syndrome, highlighting that its prevalence may be significantly reduced of about 26% in these subjects (*p* < 0.0001) [18]. 

According to the results obtained by Exenatide Study of Cardiovascular Event Lowering (EXSCEL; CT. gov identifier: NCT01144338), Exenatide reached the full CV safety but unfortunately failed to show any significant cardiovascular benefit [19], even though exenatide (10 μg) was able also to inhibit endothelial dysfunction in subjects with T2DM undergoing a meal tolerance test [20]. 

The clinical program development for “Exenatide once Weekly” (EQW) is based on multicenter, multinational, prospective, and phase 3 comparator-controlled clinical trial involving more than 5000 patients with T2DM (Diabetes Therapy Utilization: Researching Changes in HbA1c, Weight and Other Factors Through Intervention with Exenatide ONce Weekly; DURATION; CT. gov identifier: NCT00308139). The EQW program (24–30 weeks of treatment) demonstrated the ability to reduce by about 1.4% the levels of glycated hemoglobin (HbA1c), with an average of 1.94 mmol/L fasting blood glucose and 2.5 kg the body weight [21]. These significant effects were observed for up to five years in the some clinical studies, improving cardiovascular risk factors in subjects with both T2DM and the metabolic syndrome [22,23]. 

On the other hand, the therapeutic approach with exenatide Long-Acting Release (LAR) led to improved CIMT and Flow-Mediated Dilation (FMD), independently of glucometabolic status, suggesting that this therapy may represent an “add-on” to stable doses of metformin [24]. Interestingly, Exenatide significantly augmented fasting glycemia (*p* < 0.0001), HbA1c (*p* < 0.0001), waist circumference (*p* = 0.0105) and also body mass index (*p* = 0.0348), revealing unexpectedly a crucial amelioration in the lipid profile, except in triglyceride (TG) [24].

The innovative delivery version of exenatide implant (named ITCA 650) provided significant continuous subcutaneous injection of Exenatide for up to 12 months after a sub-dermal placement of a small mini-pump [25]. The FREEDOM-CVO trial (CT. gov identifier: NCT01455896) in a cardiovascular safety study demonstrated very successful results in more than 4000 patients receiving 60 micrograms per day vs. placebo for over three years [26]. This study suggests ITCA 650 as the optimal condition for once or twice-yearly sub-dermal osmotic pump for delivering continuously and consistently GLP-1 drug therapy. It is worth noting that the continuous delivery of exenatide significantly improved medication adherence, compliance, and control rates over time, crucial aspects in the management of a chronic disease like T2DM. 

Exenatide, liraglutide, and taspoglutide were able to achieve a significant reduction in total LDL-Cholesterol amount [27]. However, although at significant levels this biological effect probably is not clinically relevant, in fact, according to the Cholesterol Treatment Trialists’ collaboration, it gives only a 3% of reduction in CV events after five years [28]. A significant reduction in triglyceride levels was also shown with liraglutide 1.8 mg once daily vs. placebo [27] in patients with a mean baseline HbA1c of 8.2% (66.1 mmol/mol). Although the long-acting GLP-1 RAs showed an ability to decrease LDL-Cholesterol levels, the difference between short-acting agents and insulin did not reach statistically significant levels [29]. 

Interestingly, Semaglutide showed a CV benefit in the pre-marketing phase of the SUSTAIN-6 trial involving patients with T2DM (Cardiovascular Outcome Trial-CVOT; CT. gov identifier: NCT01720446). The trial demonstrated a very positive effect on non-fatal strokes [30]. Data of post-marketing CVOT are actually in itinere and the results will be available from a larger number of patients than those included in the previous LEADER study [31]. 

In the multicenter, randomized, double-blind, placebo-controlled, parallel-group study named Evaluation of Lixisenatide in Acute Coronary Syndrome (ELIXA; CT. gov identifier: NCT01147250), 6068 subjects with both T2DM and recent Acute Coronary Syndrome (ACS) [32] were treated with lixisenatide. After a median follow-up period of 25 months, the once daily administration of lixisenatide demonstrated safety but not superiority over placebo for the composite primary outcome (i.e., CV death, non-fatal myocardial infarction, non-fatal stroke, and/or hospitalization for unstable angina) [32,33]. 

Another long-acting GLP-1RA is Albiglutide, administered via weekly injection using a chemically structural different drug from previous marketed GLP-1 RAs [34]. HARMONY 1–8 trials (prospective, multicentre, multinational, phase 3, controlled clinical trials; CT. gov identifier: NCT00849056, NCT00849017, NCT00838903, NCT00838916, and NCT00839527) evaluated the efficacy and safety of a once-weekly administration of Albiglutide in 4838 patients with T2DM [35] showing additional significant data on cardiovascular safety and benefit within the GLP-1 analogue/agonist class. In fact, Albiglutide demonstrated its superiority with respect to placebo in patients with both T2DM and cardiovascular disease, showing a significant cardiovascular benefit (25% reduction for fatal or non-fatal myocardial infarction); unfortunately, the mechanisms involved in these effects are yet not understood [36]. 

The REWIND study on the effects of Dulaglutide on major cardiovascular events in T2DM patients (acronym of Researching Cardiovascular Events with a Weekly INcretin in Diabetes; CT. gov identifier: NCT01394952) is a double-blind, randomized, placebo-controlled study designed to assess the effects of once-weekly dulaglutide 1.5 mg on the incidence of cardiovascular outcomes [37]. Although there are previous data from a large meta-analysis considering the cardiovascular risk of dulaglutide from randomized clinical efficacy and safety trials [38], the baseline cardiovascular risks were similar between dulaglutide and comparator groups; composite major adverse cardiovascular event (MACE) endpoints of death due to cardiovascular causes, nonfatal MI, nonfatal stroke, or hospitalization for unstable angina occurred in 0.67% of patients in the dulaglutide group with respect to 1.18% in the comparator group. The relative risk of experiencing a nonfatal MI was significantly lower in the dulaglutide group versus the comparator group (*p* = 0.014). In the meantime, while awaiting the final REWIND data, this meta-analysis confirms that dulaglutide does not increase the risk of MACE in patients with T2DM based on the meta-analysis findings [38]. 

IdegLira (Degludec + Liraglutide) [39] and iGlarLixi (Glargine 100 + Lixisenatide) [40], a titratable fixed-ratio combination of insulin (Degludec or Glargine) plus GLP1-RA (Liraglutide or Lixisenatide), was found to be able to reduce the risk of hypoglycemia unawareness or hypoglycemia-associated complications, such as acute CV events. Moreover, iGlarLixi achieved significantly greater reductions in HbA1c (*p* < 0.001) and post prandial glycemia (*p* < 0.001) than that obtained with comparators, mitigating both insulin-associated weight gain and lixisenatide-associated gastrointestinal events [40]. 

It is worth noting that, in the DUAL program studies (CT. gov identifier: NCT01952145), IdegLira achieved significantly greater reductions in waist circumference (*p* = 0.0494), blood pressure (*p* = 0.0146), LDL-C (*p* = 0.0323), and tryglicerides (*p* = 0.0130) [39].

### 4.2. Sodium Glucose coTransporter-2 Inhibitors (SGLT2-is)

The inhibitors SGLT2 are carrier proteins expressed in the proximal convoluted tubule of the kidney, where they significantly contribute to the reabsorption of approximately 90% of renal glucose. Thus, SGLT2 inhibitors exert their glucose-lowering effects by reducing the renal threshold for glucose reabsorption and inducing glucose urinary excretion. A very important aspect is that SGLT2 inhibitors, due to the fact that they do not modify insulin sensitivity, are slightly associated with hypoglycemic metabolic phenomena [41].

To date, four SGLT2-inhibitors have been FDA approved: Canagliflozin, Dapagliflozin, Empagliflozin, and Ertugliflozin. 

The metabolic beneficial effects obtained by SGLT2-is are mainly linked to the ability of reducing blood pressure and decreasing extracellular volume, manifested early within the first three months of treatment [42,43]. Although in patients under dapagliflozin therapy the little changes in HDL-C, triglyceride, and LDL-C did not achieve the statistical significance, the LDL-C/HDL-C ratio was found to be consistently decreased [44]. 

Two other placebo-controlled studies of dapagliflozin (10 mg) for a duration of 12 [45] and 24 weeks [46] revealed the ability of this drug to significantly reduce small dense LDL-C (*p* < 0.005 vs. sitagliptin and *p* < 0.003 for intergroup comparison) but not the less atherogenic large LDL-C (*p* < 0.026 vs. sitagliptin, *p* < 0.671 for intergroup comparison). Interestingly, HDL-2-C (a well-known marker inversely associated with triglyceride levels and insulin resistance) was found to be significantly increased (*p* < 0.001 vs sitagliptin; *p* < 0.013 for intergroup comparison) [45]. 

Interestingly, further studies comparing linagliptin and gemigliptin vs. dapagliflozin (oriented to add-on therapy to metformin and/or sulfonylurea for 24-weeks) demonstrated a significant increase in HDL-C levels in subjects treated with SGLT-2is [46]. 

The EMPA-REG trial (acronym of Empagliflozin cardiovascular outcome events in T2DM patients; CT. gov identifier: NCT01131676.) showed that empagliflozin was able to reduce major adverse cardiovascular event MACE by 14%, with a 38% of reduction for CVD death and a reduction of 35% for heart failures [47]. 

Moreover, the CANVAS study (Canagliflozin cardioVascular Assessment Study; CT. gov identifier: NCT01032629) demonstrated that canagliflozin reduced both major adverse cardiovascular events (MACE) and hospitalization for heart failure (HHF) by 14 and 23%, respectively, even though an increased risk of amputation was reported [48]. 

The multinational observational CVD-REAL Nordic trial demonstrated a significant difference in the outcomes between the new use of SGLT-2is vs. new use of other glucose-lowering drugs, highlighting significant association among SGLT-2is with decreased risk of CVD mortality, major adverse CVD events, and HHF (*p* < 0.0001 for all conditions) [49,50]. It is worth noting that a sub-analysis of these significant data unexpectedly revealed that dapagliflozin was associated with a lower risk of MACE and HHF [50], suggesting that the well-known contrasting results about DPP4-is in CVOT are reminiscent of both the effects of thiazolidinediones on heart failure [51,52,53] and their debated uncertain/controversial outcomes in CV [53]. 

Although Dapagliflozin therapy of T2DM patients did not affect the rate of MACE when compared with placebo, it has been demonstrated that this therapeutic approach resulted in a lower rate of CV death or hospitalization for heart failure. In fact, in the DECLARE trial (Dapagliflozin and Cardiovascular Outcomes in Type 2 Diabetes; CT. gov identifier: NCT01730534) [54], based on a large cohort of 17,160 patients (including 10,186 subjects without atherosclerotic cardiovascular disease followed for a median of 4.2 years), dapagliflozin met the primary safety outcome criterions with respect to placebo concerning MACE (myocardial infarction, ischemic stroke or cardiovascular death) (*p* < 0.001). In the primary efficacy analyses (MACE and a composite of cardiovascular death or hospitalization for heart failure HHF), dapagliflozin demonstrated a lower rate of cardiovascular death or HHF (*p* = 0.005). It is worth noting that no significant differences between-group in cardiovascular death were reported. 

For what concerns the kidney patho-physiology, renal harmful events occurred in 4.3% in the dapagliflozin group and in 5.6% in the placebo group, with respect to death from any cause that occurred in 6.2% and 6.6%, respectively.

Actually, studies on renal function abnormalities with SGLT-2is and CV outcomes are in itinere (CANVAS-Renal; CT. gov identifier: NCT01032629 [55], partly included in the CANVAS Program [48]) and the results ongoing.

Some trial studies also evaluated CV death or HF-related hospitalizations in addition to the primary composite renal outcome: e.g., Evaluation of the effects of Canagliflozin on Renal and Cardiovascular Outcomes in participants with Diabetic Nephropathy (CREDENCE; CT. gov identifier: NCT02065791) trial [56]; Dapa-CKD (A Study to Evaluate the Effect of Dapagliflozin on Renal Outcomes and Cardiovascular Mortality in Patients with Chronic Kidney Disease) [57]; Dapa-HF (Study to Evaluate the Effect of Dapagliflozin on the Incidence of Worsening Heart Failure or Cardiovascular Death in Patients with Chronic Heart Failure [58]) and Empagliflozin Outcome Trial in Patients with Chronic Heart Failure with Reduced Ejection Fraction (EMPEROR-Reduced; CT. gov identifier: NCT03057977 [59]); and EMPEROR-Preserved (Empagliflozin Outcome Trial in Patients with Chronic Heart Failure with Preserved Ejection Fraction [59]). 

The EMPagliflozin compaRative effectIveness and SafEty (EMPRISE; CT. gov identifier: NCT03363464) study was aimed to assess empagliflozin’s effectiveness, safety, and healthcare utilization in routine care for five years. It investigated the risk of HHF among T2DM patients initiating empagliflozin vs. sitagliptin, a dipeptidyl peptidase-4 inhibitor (DPP-4i) [60] demonstrating that empagliflozin decreased the risk of HHF-specific by 50% and the risk of HHF-broad by 49%, independently of the doses of empagliflozin daily (10 mg or 25mg) for patients with and without baseline cardiovascular disease. Interestingly, also a comparative analyses among empagliflozin vs. the DPP4-i class, and the SGLT2i vs. DPP4-i classes revealed the same consistent findings [60].

Finally, Sotagliflozin, a novel class of inhibitor with dual target sodium-glucose co-transporter-1 (SGLT1) and -2 (SGLT2), was able to enhance the efficacy of SGLT2 inhibitors alone through the improved ability of reducing intestinal glucose absorption [61]. It is worth noting that ongoing clinical trials are documenting a significant improvement of glycaemic control in both Type 1 and Type 2 diabetes, in conjunction with smaller postprandial plasma glucose excursions, lower insulin requirements, and weight loss [61]. Thus, a cardiovascular outcome study about the Sotagliflozin effectiveness in preventing the risk of heart attack or stroke has not yet been performed.

### 4.3. Dipeptidyl Peptidase-4 Inhibitors (DPP4-is)

DDP-4 is the main enzyme involved in the biochemical pathways affecting both the degradation and inactivation of GLP-1. In fact, DPP-4 exerts proper enzymatic activity against chemotactic molecules and hormones, but showing also biological activities independently from its catalytic activity, which overall are able to modulate the intricate inflammatory, vascular, and immune processes. 

DPP-4 inhibitors, neutralizing the DPP-4 enzymatic activity, prevent the peripheral inactivation of incretins (glucose-dependent insulinotropic polypeptide and GLP-1), finally increasing the half-life and promoting the insulinotropism of GLP-1 in T2DM patients.

FDA-approved DDP4-is include sitagliptin, saxagliptin, linagliptin and alogliptin; vildagliptin has been approved from the European Medicines Agency (EMA).

On the basis of the well-known plethora of biological substrates of the enzyme DDP-4, its crucial enzymatic inhibition could lead to several bio-molecular effects, ranging from metabolic (improving glycaemic control, total cholesterol and triglyceride levels, and weight neutrality) to cardiovascular (reducing risk factors, ameliorating cardiac function and vascular repair) general improvements [1,9,10,62]. 

Although the family of DPP4-is were considered not ideal as an initial therapeutic approach, these inhitors have been nevertheless studied in patients with T2DM and HHF. However, the four trials with DPP4-i (SAVOR-TIMI 53 with saxagliptin [63], EXAMINE with alogliptin [64], TECOS with sitagliptin [65] and CARMELINA with linagliptin [66]) unfortunately failed to show any significant decrease in HHF risk in patients with T2DM. On the other hand, the SAVOR-TIMI 53 trial (Saxagliptin Assessment of Vascular Outcomes Recorded in patients with diabetes mellitus vs. placebo; CT. gov identifier: NCT01107886) reported a 27% of increased of risk for HF-related hospitalizations [63]; while, in patients with T2DM and acute coronary syndrome EXAMINE trial (EXamination of cArdiovascular outcoMes with alogliptIN vs. standard of carE; CT. gov identifier: NCT00968708) demonstrated a non-significant difference in risk with alogliptin vs. placebo [64]. 

In the TECOS study trial (Sitagliptin Cardiovascular Outcomes; CT. gov identifier: NCT00790205), a study performed on 14,671 evaluating sitagliptin or placebo treatment added to existing therapy and for a median follow-up of 3.0 years was shown only a little difference in glycated haemoglobin levels, but, more interestingly, the primary cardiovascular outcomes in patients of the sitagliptin group demonstrated that sitagliptin was considered as noninferior to placebo (*p* < 0.001) [65].

The further trial study named CARMELINA (Cardiovascular and Renal Microvascular Outcome Study with Linagliptin; CT. gov identifier: NCT01897532), recruiting 6979 patients with previous CV events and micro and/or macroalbuminuria, was found to be able to reduce eGFR with macroalbuminuria [66], reporting that the primary outcome occurred in about 12% in both the linagliptin and placebo groups (*p* < 0 .001 for noninferiority), without significant differences in the renal outcome (*p* =  0.62) [66].

Finally, the randomized, double-blind, placebo-controlled, multicenter OMNeON study (for assessing cardiovascular outcomes following omarigliptin treatment; the CT. gov identifier: NCT01703208) [67], however, demonstrated that once-weekly omarigliptin in 4202 patients with T2DM and CVD that revealed a hazard ratio (HR) of 1.00 for its primary MACE endpoint [67] was surprisingly terminated on the basis of business decisions [68]. 

## 5. Discussion and Conclusions

The CVOTs in patients with T2DM have recently focused attention on the urgent and pressing problem of HHF, which complicates and worsens the diabetic disease more frequently than myocardial infarction [69]. On the basis of several clinical trial studies, the use of glargine, degludec, sitagliptin, alogliptin, saxagliptin, lixisenatide, and once-weekly exenatide mainly showed neutral/neglectable significant effects on the major adverse cardiovascular events. Thus, these drugs (i.e., glargine, degludec, sitagliptin, alogliptin, saxagliptin, lixisenatide, and once-weekly exenatide) were found to significantly improve glycemic control and reduce microvascular complications without increasing CV risk. 

Interestingly, in a plethora of different clinical trials empagliflozin [70,71,72,73], canagliflozin [48], liraglutide [15], and semaglutide [30] have been shown to reduce CV events and deaths in very similar populations, while both empagliflozin and canagliflozin were significantly associated with a reduction in HF-related hospitalizations and deaths (*p* < 0.001) [69]. 

Additional important biological and clinical effects of empagliflozin and canagliflozin drugs were crucially linked to the significant improvement in renal function. In fact, both empagliflozin (*p* < 0.001) [72], and canagliflozin (*p* < 0.0001) [56] demonstrated the ability to slow down the well-known worsening of renal function in T2DM patients. 

Interestingly, it has also been suggested that injured renal and myocardial tissues may benefit from the access to the metabolic energy supplied by a modest increase of circulating ketone levels during treatment with the class of SGLT2 inhibitors [41,58]. 

Inside the family of DPP4 inhibitors, the linagliptin [66] shows the least impact, with no benefit beyond the glucose-lowering, except for confirming the absence of adverse effects on HF and assuring safety in the presence of renal impairment. 

Furthermore, saxagliptin and alogliptin are both associated with the unexpected increased propensity to HF [63,64,74], probably because both drugs block the cleavage of many circulating bioactive peptides and may have various downstream/bio-molecular effects not yet completely known.

Studies with the GLP-1RAs indicate that there is a significant improvement in kidney function with both liraglutide (*p* = 0.003) and semaglutide (*p* = 0.005) [15,30]. It is worth noting that the reduced progression of renal disease (i.e., significantly lower macroalbuminuria) might be secondary to the improvements in glucose metabolism and renal reabsorption, other than blood pressure improvement provided by these drugs. Interestingly, the class of molecules liraglutide and semaglutide does not generate harmful pancreatitis signals, revealing an important clinical behavior in patients with T2DM. 

Finally, the deleterious effect shown by Semaglutide on diabetic retinopathy may represent a bias and could be explained on the basis of rapid and significant improvement of glycemic control in patients with pre-existing retinopathy; this clinical behavior and biological association with a similar trend was also seen with liraglutide treatment [15,30]. 

GLP-1RAs appear to have obtained a relatively early beneficial effect on all the vasculature, physiologic effects that persist over time with the ability of protection against both MI and stroke. However, up to now, GLP-1RAs are not clearly indicated in people with T2DM and CVD.

According to the recent ADA / EASD consensus, the use of new hypoglycaemic agents, mostly SGLT-2i and GLP-1RAs, was highlighted, just before the use of metformin, also at the onset of type 2 diabetes due to its pleiotropic effects in preventing cardiovascular complications. The use of innovative therapies is marking a new era of drug therapy which is no longer based on the evaluation of the glycemic profile but above all in the tailoring management of these patients. The beneficial effects of these drugs have been widely discussed in this review; the side effects (pancreatitis, thyroid diseases, ketoacidosis, and so on) for their frequency of onset do not pose an implication in the non-use of these drugs but a laboratory and possibly instrumental monitoring associated with a very frequent clinical evaluation precisely for any adjustments of therapy.

Important CVOTs assessing CV safety of glucose-lowering medications (as depicted in Figure 1) reached their primary outcomes and confirmed previous studies that indicated no increased CV risk (as detailed in Table 2). Future data should provide additional insights into the efficacy and safety of these drugs.

## Figures and Tables

**Figure 1 jcm-09-00912-f001:**
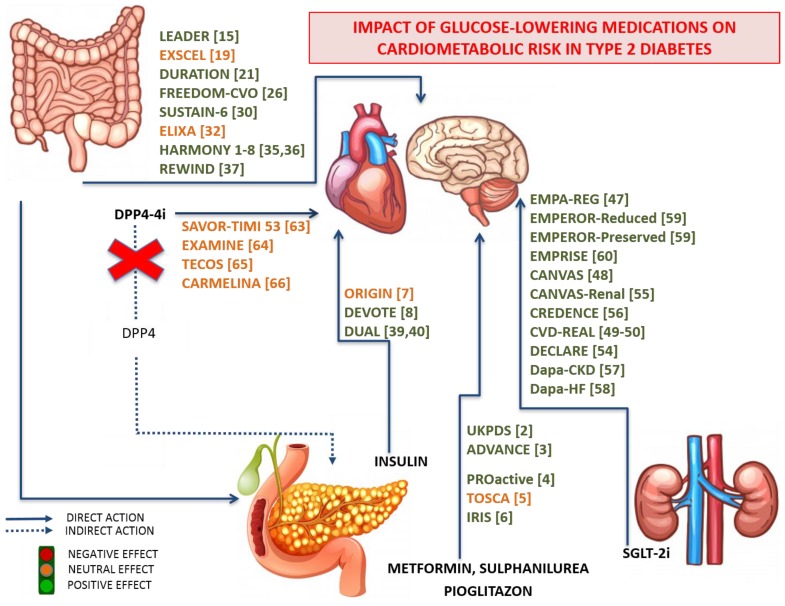
Schematic representation of the main routes, tissue biotargets, mode of actions, and effects of glucose-lowering medications, and their impact on cardiovascular risk in patients with Type 2 Diabetes. The data are obtained from both scientific literature and CardioVascular Outcome Trials, carefully assessing the cardiovascular safety of the main newest glucose-lowering medications.

**Table 1 jcm-09-00912-t001:** List of the main glucose-lowering medications and their mechanisms of action for Type 2 Diabetes and implicated in cardiovascular outcome trial (CVOT).

Class Drug	Agent	Administration	Mechanism of Action	Reference
Biguanides	Metformin	oral	↑ Insulin sensitivity by activating Adenosine Mono Phosphate-activated protein kinase (AMP-k) ↓ Hepatic glucose production	[2]
Thiazolidinediones	Pioglitazone	oral	↑ Insulin sensitivity by activation of Peroxisome Proliferator Activated Receptor gamma (PPAR-γ) ↓ Peripheral glucose uptake	[4,5,6]
Sulfonylureas	Glimepiride Gliclazide	oral oral	↑ Insulin secretion	[1,5]
Insulin	Glargine Degludec	injective	↑ Glucose disposal ↓ Hepatic glucose production	[7,8]
Dipeptidyl Peptidase-4 Inhibitors (DPP4-is)	Sitagliptin Linagliptin Omarigliptin	Oral oral oral	↓ Half-life and promoting the insulinotropism of Glucagon Like Peptide-1(GLP-1) ↑ Insulin secretion (glucose-dependent) ↓ Glucagon secretion (glucose-dependent) Enzymatic activities against chemotactic molecules and hormones modulating the intricate inflammatory, vascular and immune processes Improving glycemic control ↓ Total cholesterol and triglyceride levels Improve weight neutrality ↓ Risk factors Ameliorating cardiac function and vascular repair Block cleavage of many circulating peptides	[9,10]
Glucagon Like Peptide-1 Receptor Agonists (GLP-1RAs)	Liraglutide Exenatide Semaglutide Lixisenatide Albiglutide Dulaglutide	Injective Injective oral / injective Injective Injective Injective	↑ Insulin secretion (glucose-dependent) ↑ ß-cell proliferation ↑ Insulin biosynthesis ↓ ß-cell apoptosis ↓ Glucagon secretion (glucose-dependent) from pancreatic α-cells ↓ Rate of endogenous glucose production ↓ Gastric emptying ↑ Satiety ↓ Food intake ↓ Weight loss Improved blood pressure ↑ Control of cholesterol/dyslipidemia ↑ Low Density Lipoproteins particles oxidised (ox-LDL) ↓ Carotid Intima Media Thickness (CIMT) ↓ Flow-Mediated Dilation (FMD) ↓ Artery endothelial dysfunctions ↓ Atherosclerotic risk factors direct effects on both plaque initiation/formation and progression	[11,12,13,14,15,16,17,18,19,20,21,22,23,24,25,26,27,28,29,30,31,32,33,34,35,36,37,38,39,40]
Sodium Glucose coTransporter-2 Inhibitors (SGLT2-is)	Empagliflozin Canagliflozin Dapagliflozin	Oral oral oral	↓ Renal threshold for glucose reabsorption increasing glycosuria Modify insulin sensitivity lower insulin requirements ↓ Body weight ↓ Blood pressure ↓ Extracellular volume little changes in High Density Lipoprotein-Cholesterol (HDL-C), triglyceride, and Low Density Lipoproteins-Cholesterol (LDL-C) ↓ Small dense LDL-C	[41,42,43,44,45,46,47,48,49,50,51,52,53,54,55,56,57,58,59,60,61,62,63,64,65,66,67,68,69,70,71,72,73]

**Table 2 jcm-09-00912-t002:** Hypoglycemic drugs and cardiovascular disease (CVD)-reduction: An overview of CVOTs.

Agent	Study	Patients (N. and Type)	CVD-Reduction (Hazard Ratio HR, Confidence Interval CI, and *p*-Value)	Reference
Metformin	UK Prospective Diabetes Study (UKPDS study)	4075 overweight patients with newly diagnosed type 2 diabetes recruited in 15 centres	−32% HR (95% CI 13-47) *p* = 0.002	[2]
Pioglitazone	Prospective Pioglitazone Clinical Trial in Macrovascular Events (PROactive)	5238 patients with type 2 diabetes who had evidence of macrovascular disease	−16% HR 0.84 (95% CI 0.72–0.98) *p* = 0.027	[4]
Pioglitazone	Thiazolidinediones or Sulfonylureas Cardiovascular Accidents Intervention (TOSCA)	3028 patients with type 2 diabetes inadequately controlled with metformin monotherapy	HR 0.96 (95% CI 0.74–1.26) *p* = 0.79	[5]
Pioglitazone	Insulin Resistance Intervention After Stroke (IRIS)	3876 participants and 12% with a history of coronary artery disease	−24% HR 0.71 (95% CI 0.54–0.94) *p* = 0.02	[6]
Degludec	Efficacy and Safety of Degludec versus Glargine in Type 2 Diabetes (DEVOTE)	7637 patients with type 2 diabetes; 3818 patients with insulin degludec and 3819 patients with insulin glargine U100	HR 0.91 (95% CI 0.78–1.06) *p* = 0.21	[8]
Liraglutide	Liraglutide Effect and Action in Diabetes: Evaluation of Cardiovascular Outcome Results-A Long Term Evaluation (LEADER)	9340 patients with type 2 diabetes with a previous cardiovascular problem or chronic heart failure or at least one cardiovascular risk factor	−13.9% HR 0.87 (95% CI 0.78–0.97) *p* < 0.001	[15]
Exenatide LAR	Exenatide Study of Cardiovascular Event Lowering (EXSCEL)	14,752 patients; 10,782 had previous cardiovascular disease	−12% HR 0.91 (95% CI 0.83–1.00) *p* = 0.061	[19]
Semaglutide	Semaglutide and Cardiovascular Outcomes in Patients with Type 2 Diabetes CardioVascular Outcome Trial-CVOT (SUSTAIN-6)	3297 patients with type 2 diabetes	−6.6% HR 0.74 (95% CI 0.58–0.95) *p* < 0.001	[30]
Lixisenatide	Evaluation of Lixisenatide in Acute Coronary Syndrome (ELIXA)	6068 patients with type 2 diabetes who had had a myocardial infarction or who had been hospitalized for unstable angina within the previous 180 days	HR 1.02 (95% CI 0.89–1.17) *p* = 0.81	[32]
Albiglutide	Albiglutide and cardiovascular outcomes in patients with type 2 diabetes and cardiovascular disease (HARMONY 1-8 trials)	21,135 patients. Most patients had, or were at high risk for, cardiovascular disease.	−25% HR 0.78 (95% CI 0.68–0.90) *p* < 0.0001	[35,36]
Dulaglutide	Dulaglutide on Major Cardiovascular Events in Patients with Type 2 Diabetes: Researching Cardiovascular Events with a Weekly INcretin in Diabetes (REWIND)	9901 participants occurred in 370 sites located in 24 countries with type 2 diabetes; 31% had prior cardiovascular disease	HR 0.88 (95% CI 0.79–0.99) *p* = 0.026	[37]
Empagliflozin	Empagliflozin Cardiovascular Outcome Event Trial in Type 2 Diabetes Mellitus Patients (EMPA-REG CT)	22,830 diabetic patients	−38%; HR 0.62 (95% CI, 0.49–0.77) *p* < 0.001	[47]
Canagliflozin	Canagliflozin Cardiovascular Assessment Study (CANVAS)	10,142 participants with type 2 diabetes and high cardiovascular risk	HR 0.86 (95% CI, 0.75–0.97) *p* < 0.001	[48]
Dapagliflozin	Comparative Effectiveness of Cardiovascular Outcomes in New Users of SGLT-2 Inhibitors (CVD-REAL Nordic trial)	40,908 patients with type 2 diabetes; 23% had cardiovascular disease	HR 0.59 (95% CI, 0.49–0.72) *p* < 0.001	[49,50]
Dapagliflozin	Dapagliflozin and Cardiovascular Outcomes in Type 2 Diabetes (DECLARE)	17,160 patients, including 10,186 without atherosclerotic cardiovascular disease	HR 0.83 (95% CI, 0.73–0.95) *p* = 0.005	[54]
Sitagliptin	Sitagliptin Cardiovascular Outcomes (TECOS)	14,671 patients with type 2 diabetes and cardiovascular disease	HR 0.98 (95% CI, 0.88–1.09) *p* < 0.001	[65]
Linagliptin	Cardiovascular and Renal Microvascular Outcome Study with Linagliptin (CARMELINA)	6991 diabetic patients with high cardiovascular risk	HR 1.02 (95% CI, 0.89–1.17) *p* < 0.001	[66]
Omarigliptin	A Study to Assess Cardiovascular Outcomes Following Treatment with Omarigliptin (OMNeON study)	4202 patients with type 2 diabetes mellitus and established cardiovascular disease	HR 1.00 (95% CI 0.77–1.29) *p* = 0.77	[67,68]

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
