# Peer review of "Impact of Glucose-Lowering Medications on Cardiovascular and Metabolic Risk in Type 2 Diabetes"

_jcm, 2020, doi:10.3390/jcm9040912_

Round 1
Reviewer 1 Report
This manuscript briefly reviews the impact of glucose-lowering medications on cardiometabolic risk in type 2 diabetes. The topic is of interest, although the revision is somehow superficial.
Major comments:
English needs editing.
The first section of the review should introduce the different types of glucose-lowering drugs in the frame of their introduction in the clinical practice and their mechanism of action. In this line, authors should introduce a first table describing the agents (drugs) and their mechanisms of action before the current first table.
Authors should change “the cardiometabolic risk” with “cardiovascular risk” in the title and throughout the manuscript.
Throughout the manuscript, authors should describe the populations of the different clinical trials included, and their level of DM-associated complications.
Are there any differences in the balance benefit/risk of each glucose-lowering drugs according to the severity of DM and its complications? Authors should analyze this issue in more dept.
Minor comments:
Page 2, lines 65-67: please explain “the conventional group”.
Author Response
REVIEWER #1:
COMMENT 1:
The first section of the review should introduce the different types of glucose-lowering drugs in the frame of their introduction in the clinical practice and their mechanism of action. In this line, authors should introduce a first table describing the agents (drugs) and their mechanisms of action before the current first table.
Reply: Thanks for the suggestion that appears more appropriate and insightful for all the Readers. As you requested, we have include a detailed table citing the main mechanisms of action of hypoglycemic drugs.
COMMENT 2:
Authors should change “the cardiometabolic risk” with “cardiovascular risk” in the title and throughout the manuscript.
Reply: We agree with your suggestion, surely improving the immediate readibility of our overview. We changed "the cardiometabolic risk" in "the cardiovascular risk" in the title and throughout the ms.
COMMENT 3:
Throughout the manuscript, authors should describe the populations of the different clinical trials included, and their level of DM-associated complications.
Reply: Thanks for your insightful comment. According to your suggestion, we described in detail the patients' characteristics (number, complications, etc) in the Table 2. The patients had type 2 diabetes mellitus and were enrolled if they had a cerebro-cardiovascular event and / or risk factors predisposing to the onset of an event; some studies also evaluated a population of diabetic patients both in primary prevention (i.e. who had not yet had a cerebro-cardiovascular event) and in secondary prevention (i.e. who had already had a major cerebro-cardiovascular event).
COMMENT 4:
Are there any differences in the balance benefit/risk of each glucose-lowering drugs according to the severity of DM and its complications? Authors should analyze this issue in more dept.
Reply: Thanks for the insightful queries and useful suggestions, focused for improving the readibility and significance of our literature overview.
Accordingly, we have added comments throughout the ms and written a final comment about your issue in the discussion-conclusion section.
COMMENT 5:
Minor comments: Page 2, lines 65-67: please explain “the conventional group”.
Reply: Thanks for the useful suggestion. According to your request, we added all necessary further informations.
We would like to thanks the Reviewer 1 for the comments and suggestions focused to improve the readibility of our literature overview.

Reviewer 2 Report
Comments and Suggestions for Authors
Review of article titled: "Impact of glucose-lowering medications on cardiometabolic risk in type 2 diabetes”:
The study of the present review summarizes the results of trials that evaluated the cardiovascular safety of drugs (Glucagon Like Peptide-1 Receptor Agonists, GLP-1 RAs; Sodium Glucose coTransporter-2 inhibitors, SGLT2-is; DiPeptidyl Peptidase-4 inhibitors, DPP4-is, and other) and found them to be safe from the atherosclerotic cardiovascular standpoint.
The work appears clear and well structured. I think that Table 1 and Figure 1 are very useful and summarize the role of drugs mentioned in the text.
Corrections:
Table 1 – explain „HR” (also in the text), please.
Line 63 „Ant” – „Anti”
Introduction section: is it possible to add more references?
Author Response
REVIEWER #2:
COMMENT 1:
Table 1 – explain „HR” (also in the text), please.
Reply: Thanks for the suggestion, surely improving the immediate readibility of our ms. Accordingly, we explained the acronym HR (as Hazard Ratio) throughout the text and in the tables.
COMMENT 2:
Line 63 „Ant” – „Anti”
Reply: Thanks for helping us in the mistake identifications. We made the correction.
COMMENT 3:
Introduction section: is it possible to add more references?
Reply: Thanks for the suggestion fosued to improve the readibility of all Readers. According to your requests, we have inserted references in the introduction section.
We would like to thanks the Reviewer 2 for the suggestions focused to improve the readibility of our review ms.

Round 2
Reviewer 1 Report
This reviewer has no further comments.
This manuscript is a resubmission of an earlier submission. The following is a list of the peer review reports and author responses from that submission.